# Aroma Difference Analysis of Partridge Tea (*Mallotus oblongifolius*) with Different Drying Treatments Based on HS-SPME-GC-MS Technique

**DOI:** 10.3390/molecules28196836

**Published:** 2023-09-27

**Authors:** Xinxin Gui, Xueping Feng, Minqiang Tang, Juanling Li

**Affiliations:** Hainan Key Laboratory of Biology of Tropical Flowers and Trees Resources, Forestry Institute, Hainan University, Haikou 570228, China; guixinxin1020@163.com (X.G.); fxp151215@163.com (X.F.); tangminqiang@hainanu.edu.cn (M.T.)

**Keywords:** volatile compounds, drying method, GC-MS

## Abstract

Partridge tea has high medicinal value due to its rich content of terpenoids, phenols, flavonoids, and other related bioactive components. In order to study the best drying method for partridge tea, four treatments, including outdoor sun drying (OD), indoor shade drying (ID), hot-air drying (HAD), and low-temperature freeze-drying (LTD), were performed. The results showed that the OD and HAD treatments favored the retention of the red color of their products, while the ID and LTD treatments were more favorable for the retention of the green color. The HS-SPME-GC-MS results showed that a total of 82 compounds were identified in the four drying treatments of partridge tea, and the most abundant compounds were terpenoids (88.34–89.92%). The HAD-treated tea had the highest terpenoid content (89.92%) and high levels of flavor compounds typical of partridge tea (52.28%). OPLS-DA and PCA showed that α-copaene, β-bourbonene, caryophyllene, α-guaiene, and δ-cadinene could be considered candidate marker compounds for judging the aroma quality of partridge tea with different drying treatments. This study will not only provide a basis for processing and flavor quality control but also for spice and seasoning product development in partridge tea.

## 1. Introduction

Partridge tea (*Mallotus oblongifolius* (Miq.) Muell. Arg.) is an evergreen shrub of the genus Mallotus in the family Euphorbiaceae [1], and is mainly distributed in Hainan Province, Guangdong Province, and Guangxi Province, China [2]. Partridge tea is also an alternative tea beverage plant with strong medicinal value in Hainan Province (the southernmost province of China) [3]. Numerous biologically active components, including terpenes, phenols, and flavonoids, have been identified in partridge tea, which has anti-diabetic, anti-oxidation, anti-obesity, anti-bacterial, and anti-inflammatory properties [4,5,6,7,8].

The drying method plays a significant role in tea processing, as it directly affects the aroma, taste, color, and shelf life of tea [9]. Previously, the traditional drying methods for partridge tea were indoor shadow drying (ID) and outdoor sun drying (OD), which were time-consuming and left the tea vulnerable to environmental factors. However, nowadays, there are numerous alternative methods available for drying tea leaves. For example, two modern drying methods for tea leaves are low-temperature freeze-drying technology (LTD) and hot-air-drying technology (HAD). HAD of agricultural products is one of the most popular preservation methods because of its simplicity and low cost [10]. Moreover, previous studies had found that the polyphenol content, volatile compound content, and antioxidant capacity of hot-air-dried tea were higher than those of outdoor sun-dried tea [11]. It is well known that LTD is a drying technology that allows the product to change very little from its original state [12]. Previous research found that low-temperature freeze-dried rose tea, compared to dryer-dried, not only had improved physicochemical properties, but the color of the tea could also be retained better [13]. HAD and LTD have also been widely used in some vegetables and fruits [14], such as garlic, maoberry fruits, and lemon [15,16,17]. However, HAD and LTD have not been applied in the product processing of partridge tea.

The quality of tea is mainly determined by major factors such as color, taste, and aroma. And the formation of these factors is affected by various factors such as the processing method, tea tree variety, growing conditions, and harvesting season [18]. Particularly, aroma is the soul of tea and one of the key factors that attracts consumers [19]. However, the aroma components of tea are volatile, and the extraction process is easily affected by external conditions, so an effective aroma determination method is crucial [20]. There are many methods available for the extraction of volatile compounds, including stir bar sorptive extraction [21], solvent-assisted flavor evaporation [22], supercritical carbon dioxide extraction [23], and solid phase microextraction (SPME). Among these methods, SPME stands out as it eliminates the need for solvent extraction and instead utilizes a fiber coated with an adsorbent phase to adsorb the plant volatiles [24]. This not only allows for the adsorption of the original plant odor, but also makes the experiment easier. Gas chromatography–mass spectrometry (GC-MS) is a widely used technique for analyzing volatiles. It has a very high sensitivity and can not only obtain an overall aroma profile of the sample but also determine individual components qualitatively and quantitatively [25]. It has been widely used to identify the VOCs of many teas, such as fu brick tea [26], oolong tea [27], and dark tea [28]. This technique has the advantages of simplicity, speed, short operating time, low sample volume, no use of organic solvents, and high sensitivity [29]. Consequently, GC-MS has found extensive application in the field of food analysis, particularly in traceability analysis and variety identification [30]. In addition, data analysis on volatomics is crucial as it allows for the summarization of large and multidimensional data generated from volatomics using powerful multivariate statistical methods. Orthogonal Projections to Latent Structures Discriminant Analysis (OPLS-DA) is a supervised discriminant analysis statistical method. Partial least squares regression is used to model the relationship between metabolite expression and sample category to achieve the prediction of sample category, and it can be used for the screening of differential metabolites in metabolomics [31].

In this study, samples of partridge tea were treated with the traditional drying techniques of ID and OD as well as the current emerging drying techniques of HAD and LTD. HS-SPME-GC-MS was used to extract and characterize the volatile components of partridge tea, which will contribute to a better understanding of how the different drying methods influence the volatile compounds and enable us to scientifically elucidate the chemical basis for the aromatic qualities in partridge tea.

## 2. Results and Discussion

### 2.1. Appearance of Partridge Tea under Different Drying Methods

The appearance of the HAD- and LTD-treated tea had less leaf curling relative to the OD- and ID-treated tea, their L* values were significantly higher than those of the OD- and ID-treated tea, and the HAD and LTD treatments resulted in a brighter color (Figure 1 and Table 1). In addition, there was also a significant difference between the HAD- and LTD-treated tea; the b* value of the HAD-treated tea was significantly higher than that of the LTD-treated tea, but its a* value was significantly lower than that of the LTD-treated tea, so the HAD treatment led to better retention of the yellow appearance of the partridge tea, while the LTD treatment led to better retention of the red appearance of the partridge tea.

Cells in the leaves shrink due to uneven dehydration during slow drying, so the dried leaves curl [32], whereas the HAD-treated leaves had less leaf curling due to their high drying temperature and faster water dissipation. The reason for less curling of the leaves in the LTD-treated tea was that during the freeze-drying process, the solid water protects the basic structure and shape of the product with minimal volume change [33]. Higher temperatures during tea drying result in elevated theaflavins, theasinensins, thearubigins, and theabrownins, and most of these substances show a yellow color [34,35], so most of the partridge tea samples dried at higher temperatures showed yellow and yellow-brown colors. The lower temperature could better preserve the original color of the samples to achieve better preservation [12].

**Table 1 molecules-28-06836-t001:** Detailed descriptions of color parameters used for the classification.

Parameters	OD	ID	HAD	LTD
L*	38.95 ± 3.33 ^b^	40.56 ± 3.83 ^b^	47.28 ± 1.90 ^a^	48.14 ± 3.07 ^a^
a*	11.69 ± 2.09 ^b^	11.91 ± 1.44 ^b^	5.55 ± 2.25 ^c^	16.95 ± 2.00 ^a^
b*	5.72 ± 3.46 ^b^	5.52 ± 2.41 ^b^	13.86 ± 4.29 ^a^	1.11 ± 0.64 ^c^

Note: L* denotes luminance; a* denotes poor red-green, b* denotes poor yellow-blue [36,37]. Data are presented as mean ± standard deviation (n = 6). Mean values with different lowercase letters in the same column indicate significant differences according to the least significant difference (LSD) test (*p* < 0.05).

### 2.2. Classification of Volatile Compounds in Partridge Tea

A total of 82 compounds, including 41 alkenes, 19 aldehydes, 14 alcohols, 3 ketones, 1 alkane, and 4 esters, were identified in the four dried treatments of partridge tea via HS-SPME-GC-MS. As shown in Figure 2, there were differences in the percentages of volatile component species in different dried samples. Terpenoids were the highest in both content and quantity among the 82 compounds. The highest content of terpenoids (89.92%) was found in the HAD-treated partridge tea, along with ID (89.23%), OD (88.34%), and LTD (88.80%), indicating that the HAD-treated partridge tea was able to retain more flavor compounds. Terpenes are natural compounds found primarily in the plant world [38]. Most terpenes have antibacterial, anticancer, anti-inflammatory, and other biological activities [39], and are widely used in the pharmaceutical, food, and cosmetic industries [40]. Therefore, the HAD treatment was favorable for improving the nutritional and medicinal value of partridge tea.

### 2.3. Identification of Volatile Compounds in Partridge Tea

Among the 82 compounds identified (Table 2, Figure 3), 55 components were significantly different between the four drying treatments (*p* < 0.05). In this study, the content of α-copaene, β-bourbonene, caryophyllene, α-guaiene, α-caryophyllene, and δ-cadinene was found to be higher and all were terpenes, and are typical partridge tea flavor compounds according to previous studies [41]. These flavor compounds were highest in the HAD-treated (52.28%), followed by the OD-treated (49.64%), and then, the LTD-treated (45.34%) and ID-treated (43.78%) tea. All of these typical flavor compounds of partridge tea have a chemical structure of C_15_H_24_ and are sesquiterpenoids, which have excellent anti-tumor, anti-inflammatory, and antibacterial properties [42]. Particularly, caryophyllene and β-bourbonene have been confirmed to have good anticancer activity, and caryophyllene shows a selective antiproliferative effect on colorectal cancer cells [43,44]. β-bourbonene was shown to inhibit the proliferation of prostate cancer PC-0M cells [45]. Our results show that the HAD drying method was more favorable for the retention of terpenoids, and the HAD-treated partridge tea had better biological activities such as anticancer and antivirus.

### 2.4. Orthogonal Partial Least Squares Discriminant Analysis

PCA and OPLS-DA were used to downgrade and screen the differential components of partridge tea treated with different drying methods, and then, classify the chemotypes according to the dominant compounds among the different drying treatments. PCA showed that PC1 and PC2 accounted for 51.6% of the total variability of the model. As shown in Figure 3B (R2X = 0.691, R2Y = 0.977, Q2 = 0.904), the orthogonal partial least squares (OPLS-DA) analysis was performed to obtain the variable importance of projection (VIP) values (important variable in the projection).

Based on the results of OPLS-DA, volatile compounds with VIP values > 2 were selected, and the S-plot showed 12 compounds with significant differences among the four different drying treatments of partridge tea (data point triangle on the plot) (Figure 4C–H), including 2-methyl-2-butenal, (e)-2-hexenal, α-copaene, β-bourbonene, β-maaliene, caryophyllene, isogermacrene d, α-guaiene, (e)-β-famesene, eremophilene, (+)-ledene, and δ-cadinene. These 12 compounds were significantly different in the four different drying treatments and could be considered candidate marker compounds for judging the aroma quality of partridge tea with different drying treatments, and provide a reference for the use of partridge tea for making fragrances and aromatics as well as for the control of partridge tea product quality.

## 3. Materials and Methods

### 3.1. Sample Preparation

The experimental samples were collected on 12 March 2023 at the Wild Partridge Tea Germplasm Resource Nursery of Hainan University, and we selected partridge tea plants with relatively consistent leaf color and growth conditions as the sample collection objects. The four groups of samples were collected from the third fresh leaves under the buds, with three replicates in each group, and each replicate was collected with about 50 g for the subsequent drying process (Table 3).

### 3.2. Volatile Extraction via HS-SPME Procedure

The samples obtained from different drying treatments were crushed and 0.5 g added to a headspace flask, and the headspace flask was placed in a water bath at 65 °C for extraction. Before extraction, the solid-phase microextraction head was aged at the gas chromatography inlet at 250 °C for 10 min. After a period of sample equilibration (5 min), the needle of the SPME device was inserted into the vial through the septum, and the fiber was exposed to the vial headspace for 40 min. After that time, the fiber was retracted into the needle assembly, removed from the vial, transferred to the injection port of the GC unit, and immediately desorbed.

### 3.3. Separation of Volatile Organic Compounds via GC-MS

An Agilent 7890A GC system, coupled with a Tekmar 5975C mass spectrometer (Agilent/Tekmar, Santa Clara, CA, USA), was applied in the present study. An Agilent HP-5MS capillary column (30 m × 0.25 mm, 0.25 μm) was employed to separate the volatiles. Helium (>99.99%) was used as the carrier gas at a constant flow rate of 1 mL/min. The GC oven temperature was maintained at 40 °C for 4 min, ramped at a rate of 5 °C/min to 160 °C, and ramped at a rate of 10 °C/min to 260 °C. The mass spectrometer conditions were as follows: electron ionization energy of 70 eV, ion source temperature of 230 °C, interface temperature of 280 °C, and mass scan range of 50–550 *m*/*z*.

### 3.4. Identification and Relative Quantification of Volatile Organic Compounds via GC-MS

The obtained chromatograms were analyzed by determining the peak areas, retention times, spectra, and base peaks on the chromatograms, and then, each peak was examined by referring to the characteristic mass spectra of the compounds listed on the National Institute of Standards and Technology (NIST14. L). Compounds were identified based on their similarity indexes (>80%) as well as retention indexes. The relative contents of each compound in the volatile compounds of tea samples were obtained via the area normalization method.

### 3.5. Determination of Color Differences Using a Colorimeter

A portable colorimeter (LS173B, Shenzhen Linshang Technology Co., Ltd., Shenzhen, China) was utilized to measure the color differences of partridge tea leaves subjected to four different drying treatments. Six dried leaves were selected for each treatment group. Before taking measurements, the colorimeter was calibrated using black and white references. The leaves were then placed tightly against the test holes, completely covering them, and measurements were subsequently recorded.

### 3.6. Statistical Analysis

Results were expressed as the mean ± SD (n = 3) for each analysis. Statistical analysis was performed via one-way analysis of variance (ANOVA) using IBM SPSS 26.0 (SPSS Inc., Chicago, IL, USA). Differences between the groups were evaluated via the Duncan test for multiple comparisons. Origin 2018c (OriginLab, Northampton, MA, USA) was used for graphic drawing. *p* < 0.05 was considered to indicate a statistically significant difference. SIMCA 14.1 (Sartorius AG, Göttingen, Germany) software was used for orthogonal partial least squares discriminant analysis (OPLS-DA).

## 4. Conclusions

In this study, HS-SPME-GC-MS and a colorimeter were used to identify volatile compounds and the appearance quality of partridge tea using four drying methods. It was found that the drying methods had a greater influence on the appearance of the tea leaves. Cold drying methods (ID and LTD) had a greater effect on the retention of leaf color, while the two relatively hot-air drying methods (OD and HAD) had a poorer effect on the retention of leaf color, with an overall darker brown color. In addition, a total of 82 compounds were identified in the different drying treatments for partridge tea. Among them, terpenoids were the most important class of compounds contributing to the aroma composition of partridge tea. And HAD had the highest terpenoid content among the four drying treatments. Meanwhile, we found that the five active ingredients were α-copaene, β-bourbonene, caryophyllene, α-guaiene, and δ-cadinene, and the HAD treatment had a relatively good retention effect on these compounds, which could be used as marker compounds to evaluate the quality of partridge tea. This study investigated the effect of different drying methods on the volatile compounds and the appearance quality of partridge tea, which provides a basis for the selection of processing methods for partridge tea. Finally, this study confirmed essential volatile compounds via a chemometric method in partridge tea, which provides a basis for the application of partridge tea in spices and seasonings.

## Figures and Tables

**Figure 1 molecules-28-06836-f001:**
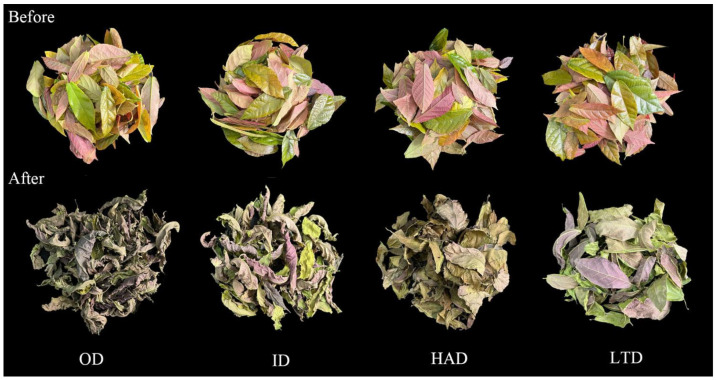
Differences in appearance before and after different drying treatments of partridge tea.

**Figure 2 molecules-28-06836-f002:**
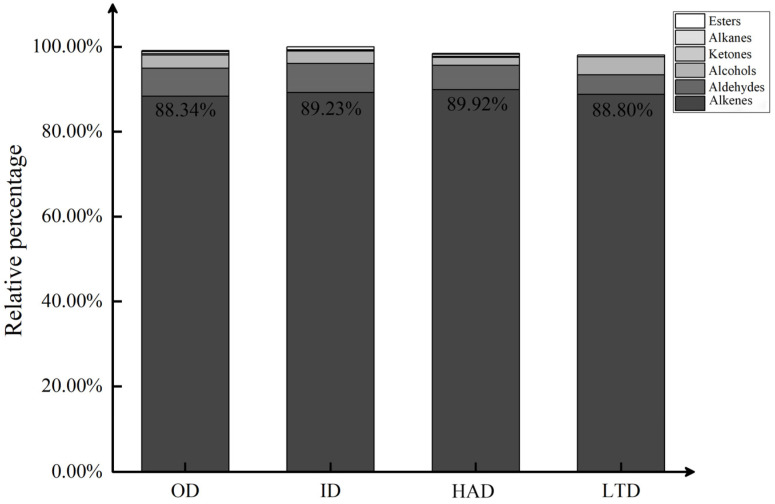
Histogram of relative content of different categories of volatile substances.

**Figure 3 molecules-28-06836-f003:**
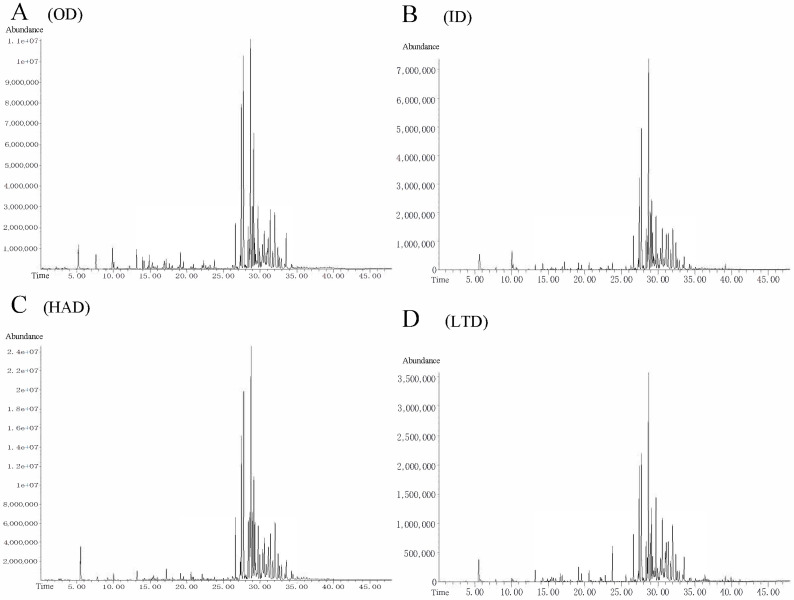
GC chromatograms of changes in OD (**A**), ID (**B**), HAD (**C**), and LTD (**D**).

**Figure 4 molecules-28-06836-f004:**
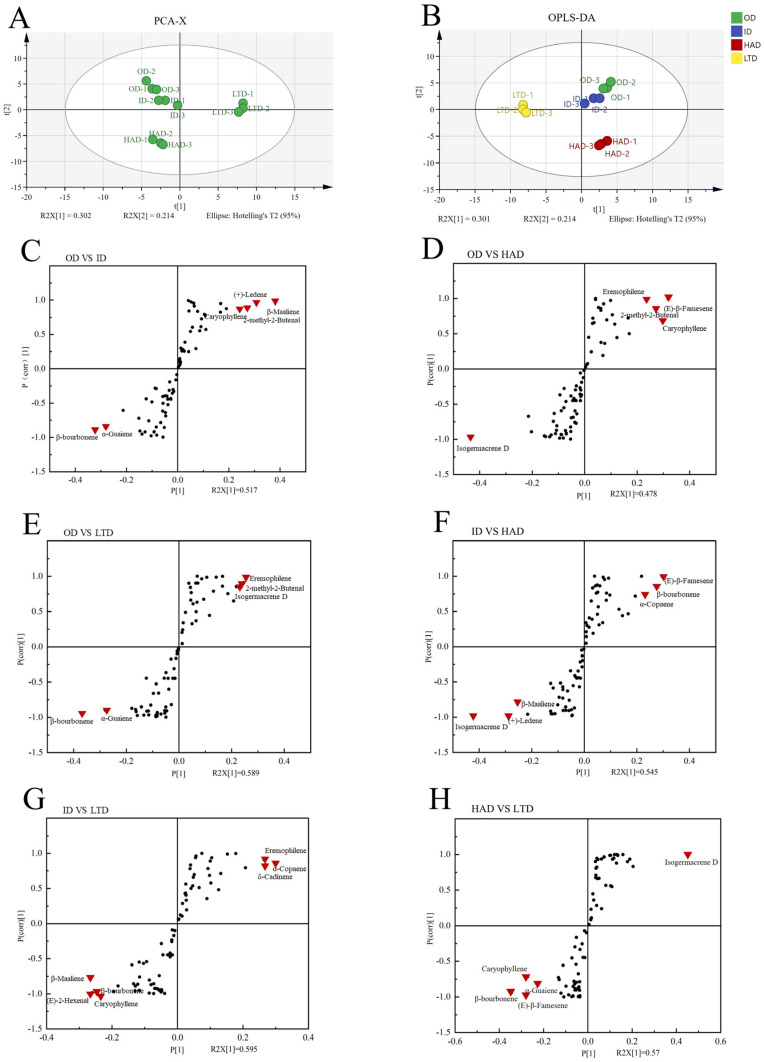
Principal component analysis (PCA) (**A**), orthogonal partial least squares (OPLS-DA) (**B**). The S-plot (**C**–**H**) shows the six different comparative models, and the candidate marker compounds within each model (data point triangle on the plot).

**Table 2 molecules-28-06836-t002:** Relative percentage content of volatiles of different drying treatments for partridge tea.

No.	Compounds	Formula	RT	RI	OD (%)	ID (%)	HAD (%)	LTD (%)
**1**	2-Methyl-2-butenal	C_5_H_8_O	5.24	737	0.42 ± 0.73 ^b^	2.18 ± 0.21 ^a^	2.58 ± 0.96 ^a^	2.22 ± 0.33 ^a^
**2**	cis-2-Pentenol	C_5_H_10_O	6.75	769	0.02 ± 0.03 ^a^	0.03 ± 0.05 ^a^	0.05 ± 0.02 ^a^	Nd ^a^
**3**	Hexanal	C_6_H_12_O	7.62	800	0.94 ± 0.14 ^a^	0.42 ± 0.15 ^b^	0.33 ± 0.09 ^bc^	0.19 ± 0.00 ^c^
**4**	Furfural	C_5_H_4_O_2_	9.21	835	Nd ^b^	Nd ^b^	0.25 ± 0.05 ^a^	Nd ^b^
**5**	(E)-2-Hexenal	C_6_H_10_O	9.88	850	1.06 ± 0.31 ^b^	1.94 ± 0.29 ^a^	0.58 ± 0.20 ^c^	0.21 ± 0.02 ^c^
**6**	3-Hexen-1-ol	C_6_H_12_O	10.18	858	Nd ^c^	Nd ^c^	0.08 ± 0.04 ^b^	0.14 ± 0.00 ^a^
**7**	(E)-2-Hexen-1-ol	C_6_H_12_O	10.50	861	0.09 ± 0.03 ^b^	0.19 ± 0.03 ^a^	0.03 ± 0.06 ^bc^	Nd ^c^
**8**	Sorbaldehyde	C_6_H_8_O	12.19	913	0.23 ± 0.04 ^a^	0.05 ± 0.09 ^b^	Nd ^b^	Nd ^b^
**9**	α-Pinene	C_10_H_16_	13.16	939	0.63 ± 0.36 ^a^	0.79 ± 0.42 ^a^	0.46 ± 0.06 ^a^	0.88 ± 0.13 ^a^
**10**	(Z)-2-Heptenal	C_7_H_12_O	13.97	958	0.20 ± 0.35 ^a^	0.02 ± 0.04 ^a^	0.10 ± 0.04 ^a^	Nd ^a^
**11**	(E)-2-Heptenal	C_7_H_12_O	13.98	962	0.49 ± 0.09 ^a^	0.05 ± 0.08 ^b^	0.03 ± 0.06 ^b^	0.09 ± 0.01 ^b^
**12**	Benzaldehyde	C_7_H_6_O	14.17	966	0.53 ± 0.21 ^a^	0.56 ± 0.10 ^a^	0.19 ± 0.04 ^b^	0.52 ± 0.02 ^a^
**13**	β-Pinene	C_10_H_16_	14.84	974	Nd ^b^	0.09 ± 0.03 ^a^	0.01 ± 0.02 ^b^	Nd ^b^
**14**	Matsutake alcohol	C_8_H_16_O	14.88	976	0.54 ± 0.12 ^a^	0.16 ± 0.08 ^b^	0.11 ± 0.07 ^b^	Nd ^b^
**15**	Prenylacetone	C_8_H_14_O	15.10	980	0.09 ± 0.08 ^a^	0.02 ± 0.03 ^a^	0.08 ± 0.02 ^a^	0.02 ± 0.03 ^a^
**16**	(E,E)-2,4-Heptadienal	C_7_H_10_O	15.97	1007	0.21 ± 0.08 ^a^	0.15 ± 0.01 ^a^	0.18 ± 0.03 ^a^	0.23 ± 0.02 ^a^
**17**	2-Ethylhexanol	C_8_H_18_O	16.65	1015	Nd ^b^	Nd ^b^	Nd ^b^	0.59 ± 0.09 ^a^
**18**	Limonene	C_10_H_16_	16.69	1030	0.07 ± 0.03 ^a^	0.07 ± 0.02 ^a^	0.05 ± 0.01 ^a^	Nd ^b^
**19**	Benzenemethanol	C_7_H_8_O	16.89	1033	0.25 ± 0.05 ^b^	0.28 ± 0.09 ^b^	0.11 ± 0.01 ^c^	0.58 ± 0.01 ^a^
**20**	Benzeneacetaldehyde	C_8_H_8_O	17.18	1042	0.51 ± 0.17 ^a^	0.44 ± 0.10 ^a^	0.58 ± 0.14 ^a^	0.08 ± 0.01 ^b^
**21**	β-Ocimene	C_10_H_16_	17.32	1044	Nd ^b^	0.04 ± 0.01 ^a^	0.04 ± 0.01 ^a^	Nd ^b^
**22**	(E)-2-Octenal	C_8_H_14_O	17.61	1056	0.23 ± 0.03 ^a^	0.08 ± 0.03 ^b^	0.06 ± 0.01 ^b^	Nd ^c^
**23**	3,5-Octadien-2-one	C_8_H_12_O	18.04	1081	0.20 ± 0.04 ^a^	0.13 ± 0.01 ^b^	0.17 ± 0.02 ^ab^	Nd ^c^
**24**	Linalool	C_10_H_18_O	19.05	1098	0.16 ± 0.03 ^ab^	0.20 ± 0.09 ^a^	0.08 ± 0.02 ^bc^	0.02 ± 0.03 ^c^
**25**	Nonanal	C_9_H_18_O	19.14	1102	0.74 ± 0.12 ^ab^	0.60 ± 0.17 ^bc^	0.41 ± 0.10 ^c^	0.87 ± 0.09 ^a^
**26**	Benzeneethanol	C_8_H_10_O	19.55	1114	0.36 ± 0.07 ^b^	0.23 ± 0.12 ^bc^	0.21 ± 0.04 ^c^	0.52 ± 0.03 ^a^
**27**	Cucumber aldehyde	C_9_H_14_O	20.74	1151	0.03 ± 0.06 ^b^	Nd ^b^	0.15 ± 0.02 ^a^	Nd ^b^
**28**	(E)-2-Nonenal	C_9_H_16_O	20.91	1161	0.23 ± 0.05 ^a^	0.03 ± 0.06 ^c^	0.15 ± 0.02 ^b^	Nd ^c^
**29**	1-Dodecene	C_12_H_24_	21.85	1192	0.26 ± 0.00 ^a^	0.14 ± 0.09 ^a^	0.20 ± 0.10 ^a^	0.23 ± 0.02 ^a^
**30**	Myrtenol	C_10_H_16_O	22.22	1194	0.16 ± 0.04 ^a^	0.14 ± 0.04 ^a^	0.03 ± 0.00 ^b^	0.16 ± 0.04 ^a^
**31**	Decanal	C_10_H_20_O	22.31	1200	0.40 ± 0.04 ^a^	0.16 ± 0.04 ^b^	0.07 ± 0.12 ^b^	0.19 ± 0.02 ^b^
**32**	(E,E)-2,4-Nonadienal	C_9_H_14_O	22.60	1214	0.21 ± 0.08 ^a^	0.01 ± 0.02 ^b^	Nd ^b^	Nd ^b^
**33**	β-Cyclocitral	C_10_H_16_O	22.92	1218	0.11 ± 0.02 ^ab^	0.12 ± 0.03 ^a^	0.08 ± 0.01 ^b^	0.02 ± 0.03 ^c^
**34**	cis-3-Hexenyl valerate	C_11_H_20_O_2_	23.18	1235	0.21 ± 0.06 ^b^	0.47 ± 0.20 ^a^	0.11 ± 0.06 ^b^	Nd ^b^
**35**	Butanoic acid, 3-methyl-, hexyl ester	C_11_H_22_O_2_	23.33	1244	Nd ^b^	0.06 ± 0.02 ^a^	Nd ^b^	Nd ^b^
**36**	Geraniol	C_10_H_18_O	23.76	1250	0.37 ± 0.06 ^b^	0.67 ± 0.19 ^b^	0.17 ± 0.03 ^b^	1.30 ± 0.48 ^a^
**37**	(E)-2-Decenal	C_10_H_18_O	23.98	1260	0.07 ± 0.01 ^a^	Nd ^b^	Nd ^b^	Nd ^b^
**38**	α-Citral	C_10_H_16_O	24.24	1269	0.01 ± 0.02 ^b^	0.07 ± 0.01 ^a^	Nd ^b^	Nd ^b^
**39**	δ-Elemene	C_15_H_24_	26.29	1316	0.25 ± 0.04 ^b^	0.41 ± 0.08 ^a^	0.20 ± 0.01 ^b^	0.39 ± 0.04 ^a^
**40**	α-Cubebene	C_15_H_24_	26.63	1348	2.33 ± 0.07 ^a^	2.34 ± 0.23 ^a^	2.37 ± 0.47 ^a^	2.55 ± 0.22 ^a^
**41**	3-Methyl-tridecane	C_14_H_30_	26.94	1371	0.02 ± 0.04 ^a^	0.03 ± 0.06 ^a^	0.13 ± 0.04 ^a^	0.04 ± 0.08 ^a^
**42**	Ylangene	C_15_H_24_	27.29	1372	0.90 ± 0.05 ^b^	0.86 ± 0.05 ^b^	1.04 ± 0.02 ^a^	0.88 ± 0.11 ^b^
**43**	α-Copaene	C_15_H_24_	27.44	1376	6.28 ± 1.61 ^a^	5.68 ± 0.73 ^a^	7.71 ± 1.41 ^a^	8.22 ± 1.14 ^a^
**44**	β-Bourbonene	C_15_H_24_	27.72	1382	11.50 ± 1.11 ^a^	8.96 ± 0.14 ^b^	11.39 ± 1.23 ^a^	7.34 ± 0.36 ^b^
**45**	Cyperene	C_15_H_24_	28.16	1390	0.11 ± 0.01 ^a^	0.13 ± 0.07 ^a^	0.15 ± 0.02 ^a^	0.03 ± 0.05 ^b^
**46**	α-Gurjenene	C_15_H_24_	28.37	1401	0.77 ± 1.33 ^a^	0.02 ± 0.03 ^a^	1.39 ± 2.41 ^a^	0.87 ± 1.30 ^a^
**47**	β-Maaliene	C_15_H_24_	28.38	1413	Nd ^b^	3.14 ± 0.27 ^a^	0.84 ± 1.45 ^b^	0.89 ± 1.54 ^b^
**48**	α-Santalene	C_15_H_24_	28.55	1420	1.73 ± 0.15 ^a^	2.38 ± 0.11 ^a^	2.67 ± 0.76 ^a^	1.95 ± 0.54 ^a^
**49**	Caryophyllene	C_15_H_24_	28.71	1428	13.82 ± 0.69 ^b^	15.28 ± 0.07 ^ab^	16.98 ± 2.77 ^a^	13.66 ± 0.27 ^b^
**50**	Calarene	C_15_H_24_	28.93	1434	4.09 ± 0.32 ^a^	4.38 ± 0.08 ^a^	3.49 ± 1.26 ^a^	3.39 ± 0.19 ^a^
**51**	Isogermacrene D	C_15_H_24_	28.94	1439	4.79 ± 0.77 ^b^	5.17 ± 0.75 ^b^	Nd ^c^	6.53 ± 0.52 ^a^
**52**	α-Guaiene	C_15_H_24_	29.12	1440	7.06 ± 1.07 ^a^	5.03 ± 0.13 ^bc^	6.52 ± 1.24 ^ab^	4.49 ± 0.25 ^c^
**53**	Aromandendrene	C_15_H_24_	29.26	1441	2.07 ± 0.31 ^b^	2.80 ± 0.31 ^a^	2.58 ± 0.44 ^ab^	1.87 ± 0.30 ^b^
**54**	α-Caryophyllene	C_15_H_24_	29.72	1452	6.02 ± 1.55 ^a^	4.39 ± 1.00 ^a^	4.33 ± 0.39 ^a^	5.01 ± 0.81 ^a^
**55**	Alloaromadendrene	C_15_H_24_	29.94	1454	1.64 ± 0.35 ^a^	1.67 ± 0.25 ^a^	1.31 ± 0.17 ^a^	1.22 ± 0.22 ^a^
**56**	β-Copaene	C_15_H_24_	30.10	1459	0.98 ± 0.14 ^a^	0.89 ± 0.13 ^a^	0.41 ± 0.02 ^b^	0.97 ± 0.01 ^a^
**57**	Germacrene d	C_15_H_24_	30.12	1461	0.36 ± 0.12 ^a^	0.29 ± 0.05 ^a^	0.29 ± 0.06 ^a^	0.35 ± 0.13 ^a^
**58**	γ-Gurjunene	C_15_H_24_	30.26	1471	0.11 ± 0.20 ^b^	0.44 ± 0.12 ^a^	Nd ^b^	Nd ^b^
**59**	γ-Muurolene	C_15_H_24_	30.34	1474	2.45 ± 0.48 ^a^	2.34 ± 0.13 ^a^	2.70 ± 0.26 ^a^	2.78 ± 0.54 ^a^
**60**	(E)-β-Famesene	C_15_H_24_	30.58	1476	Nd ^b^	Nd ^b^	2.51 ± 0.21 ^a^	Nd ^b^
**61**	β-Selinene	C_15_H_24_	30.81	1479	1.35 ± 0.23 ^a^	1.38 ± 0.20 ^a^	1.07 ± 0.29 ^a^	0.94 ± 0.06 ^a^
**62**	Eremophilene	C_15_H_24_	31.02	1486	Nd ^b^	Nd ^b^	1.35 ± 0.08 ^a^	1.95 ± 0.77 ^a^
**63**	(+)-Ledene	C_15_H_24_	31.11	1489	2.87 ± 0.42 ^b^	5.05 ± 0.34 ^a^	2.59 ± 0.19 ^b^	4.20 ± 0.92 ^a^
**64**	α-Bulnesene	C_15_H_24_	31.40	1503	4.03 ± 0.21 ^a^	3.34 ± 0.54 ^a^	3.50 ± 0.52 ^a^	3.14 ± 0.34 ^a^
**65**	γ-Cadinene	C_15_H_24_	31.72	1507	1.21 ± 0.33 ^b^	1.50 ± 0.19 ^ab^	1.24 ± 0.09 ^b^	1.84 ± 0.27 ^a^
**66**	δ-Cadinene	C_15_H_24_	32.00	1508	4.97 ± 0.82 ^b^	4.45 ± 1.25 ^b^	5.37 ± 0.32 ^ab^	6.61 ± 0.25 ^a^
**67**	Cubenene	C_15_H_24_	32.43	1512	1.55 ± 0.05 ^a^	2.01 ± 0.57 ^a^	1.56 ± 0.30 ^a^	1.94 ± 0.36 ^a^
**68**	Selina-4(15),7(11)-diene	C_15_H_24_	32.62	1544	0.85 ± 0.04 ^a^	0.95 ± 0.13 ^a^	1.11 ± 0.22 ^a^	0.94 ± 0.05 ^a^
**69**	α-Calacorene	C_15_H_20_	32.88	1546	Nd ^b^	Nd ^b^	Nd ^b^	0.80 ± 0.04 ^a^
**70**	Selina-3,7(11)-diene	C_15_H_24_	32.90	1550	0.75 ± 0.08 ^ab^	0.81 ± 0.17 ^ab^	1.11 ± 0.25 ^a^	0.34 ± 0.58 ^b^
**71**	E-Nerolidol	C_15_H_26_O	33.37	1551	0.48 ± 0.29 ^a^	0.48 ± 0.06 ^a^	0.47 ± 0.07 ^a^	0.45 ± 0.10 ^a^
**72**	Germacrene B	C_15_H_24_	33.57	1556	2.44 ± 0.22 ^a^	1.98 ± 0.68 ^ab^	1.30 ± 0.45 ^b^	1.36 ± 0.47 ^b^
**73**	(+)-Spathulenol	C_15_H_24_O	34.27	1557	0.51 ± 0.27 ^a^	0.38 ± 0.05 ^a^	0.41 ± 0.09 ^a^	0.25 ± 0.01 ^a^
**74**	Caryophyllene oxide	C_15_H_24_O	34.47	1561	0.48 ± 0.19 ^a^	0.11 ± 0.20 ^bc^	0.38 ± 0.01 ^ab^	Nd ^c^
**75**	Ledol	C_15_H_26_O	35.09	1565	0.14 ± 0.04 ^a^	0.15 ± 0.04 ^a^	0.12 ± 0.03 ^a^	0.18 ± 0.04 ^a^
**76**	α-Corocalene	C_15_H_20_	35.49	1629	0.02 ± 0.02 ^b^	0.02 ± 0.03 ^b^	0.01 ± 0.01 ^b^	0.07 ± 0.01 ^a^
**77**	Cadalene	C_15_H_18_	36.80	1674	0.03 ± 0.03 ^b^	0.01 ± 0.02 ^b^	0.02 ± 0.02 ^b^	0.09 ± 0.01 ^a^
**78**	Guaiazulene	C_15_H_18_	38.77	1772	0.05 ± 0.05 ^a^	0.03 ± 0.03 ^a^	0.03 ± 0.01 ^a^	0.07 ± 0.02 ^a^
**79**	Isopropyl myristate	C_17_H_34_O_2_	39.18	1823	0.02 ± 0.04 ^b^	0.15 ± 0.13 ^ab^	0.07 ± 0.02 ^b^	0.25 ± 0.03 ^a^
**80**	Neophytadiene	C_20_H_38_	39.41	1837	Nd ^a^	Nd ^a^	0.04 ± 0.00 ^a^	Nd ^a^
**81**	Hexahydrofarnesyl acetone	C_18_H_36_O	39.51	1847	0.07 ± 0.06 ^a^	Nd ^a^	Nd ^a^	0.04 ± 0.06 ^a^
**82**	Diisobutyl phthalate	C_16_H_22_O_4_	39.92	1868	Nd ^a^	Nd ^a^	Nd ^a^	0.09 ± 0.09 ^a^

Note: OD stands for outdoor sun drying; ID stands for indoor shade drying; HAD stands for oven hot-air drying; LTD stands for low-temperature freeze-drying. Nd stands for not detected. Results are expressed as mean ± standard error. Different letters in the same row indicate significant differences at *p* < 0.05.

**Table 3 molecules-28-06836-t003:** Basic information on tea samples.

Sample Name	Year	Drying Method	Temperature	Location
OD	2023	Outdoor sun drying	30–35 °C	Hainan, China
ID	2023	Indoor shade drying	25 ± 2 °C	
HAD	2023	Hot-air drying	70 °C	
LTD	2023	Low-temperature drying	1. Temperature down to −25 °C lyophilization for 4 h; 2. temperature down to −30 °C lyophilization for 10 h; 3. temperature up to −25 °C lyophilization for 2 h; 4. temperature up to −5 °C lyophilization for 5 h; 5. temperature up to 15 °C drying for 5 h; 6. temperature up to 30 °C drying for 6 h, drying completed.	

## Data Availability

Not applicable.

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
