# Peer review of "Aroma Difference Analysis of Partridge Tea (*Mallotus oblongifolius*) with Different Drying Treatments Based on HS-SPME-GC-MS Technique"

_molecules, 2023, doi:10.3390/molecules28196836_

Round 1
Reviewer 1 Report
“Higher temperatures during tea drying result in elevated theaflflavins, theasinensins, thearubigins, and theabrownins, and most of these substances showed a yellow color so most of the partridge tea samples dried at higher temperatures showed yellow and yellow-brown color.” The partridge tea is not based on Camellia species, so the color formation mechanism of tea (Camellia sp.) is far away from that of partridge. Have authors determine the main compounds’ content of partridge tea? Does it contain the flavan-3ols?
About the color changing, have authors determined the contents of chlorophyll of fours samples?
“….. 41 terpenoids, 19 aldehydes, 13 alcohols, 3 ketones, 2 alkanes, and 4 esters……”, it is a little weird that terpenoids are classified together with alcohols. Because terpenoids could be alcohols, alkanes, even esters” I am confused about the classification of volatiles as this way.
Reviewer 2 Report
Paper needs to be reviewed and improved. There are major issues is the M&M section and the R&D section. More detail and a better justification and explanation of results should be provided.
1. More detail is needed in the sample collection process, the amount of sample collected, etc.
2. In table 1 some T show standard deviations, other show ranges and for the LTD the explanation is too long.
3. Section 3.4 needs more detail and the manufacturers of the libraries used should be presented.
4. Section 3.5 needs to provide the manufacturers and information of the software used.
5. Conclusions talk about colorimetry but nothing is mentioned in the M&M section.
6. Line 121, the sentence HAD-treated partridge tea had better biological activities...... , how is this proven in this study, please review and present references to support this affirmation.
7. Explain in M&M, how was RI calculated.
8. Caption in figure 3 says flow diagram, but the images are actually GC chromatograms, change.
9. Figure 4 caption and images are not clear and explanation is vague.
Article needs to be edited for major english grammar mistakes.
Reviewer 3 Report
Ø In the title and all manuscript, the scientific name of Partridge Tea must be included.
Ø Between lines 54-60 on HS-SPME-GC-MS references on this analytical technique are needed. Some examples of timely references that I recommend:
- Volatilomics for food quality and authentication. DOI: https://doi.org/10.1016/j.cofs.2019.10.003
- Volatilomics of Natural Products: Whispers from Nature. DOI:10.5772/intechopen.97228
In addition, a more detailed explanation of the characteristics of the technique, to give more context as to why to use this technique instead of solvent-liquid extraction.
Ø In the introduction section, the authors should provide a relevant explanation of the multivariate statistics (OPLS-DA)) used in the study to give readers an appropriate context.
Ø Although Partridge Tea is a locally consumed product in the Asian region, the authors' literature review should be more global. Authors should not exclusively cite from mostly Asian authors 31of 37 references. HS-SPME-GC-MS is used worldwide by scientists working on VOCs.
Ø It is understandable that the authors' objective is to compare the drying methods for Partridge Tea; however, why did they not optimize the chromatographic separation by modifying the heating ramp between minutes 25 to 35?
A higher chromatographic resolution would allow a more reliable identification in your study. In particular, because the major compounds (> 88%) in your sample correspond to the terpenoids that elute in these minutes of the chromatogram.
Ø The name of section 3.3 should be changed to: Separation of volatile organic compounds by GC-MS.
Ø The name of section 3.4 should be changed to: identification and relative quantification of volatile organic compounds by GC-MS. In this research we are not using calibration curves or an internal standard, therefore we cannot speak of quantification or semi-quantification.
Ø The authors mention that the identification of VOCs in section 3.4 was performed with authentic standards, but in the manuscript, there is no section on reagents and standards where they list which standards were used in this research. Please, clarify this information.
Ø Verify if the word theaflflavins on line 82 really corresponds to theaflavin.
Ø In lines 102-103 when it is mentioned that terpenoids are compounds with high nutritional and medicinal values, a more detailed explanation should be given and not as general as the one presented in the current version of the manuscript.
Ø In the conclusions section, on line 202, change the word and to finally.
Not comments
Round 2
Reviewer 1 Report
no comments,accept it as its revised version
Author Response
Dear reviewer,
Thank you very much for your comments and professional advice. These opinions help to improve academic rigor of our article. Meanwhile, we have made appropriate changes for the language quality of the manuscript. We hope that our work can be improved again and that we will be able to meet the publication requirements of your journal. Thank you again for taking time out of your busy schedule to review our manuscript.
Thank you very much for your attention and time.
Your sincerely.
Xinxin Gui
Reviewer 2 Report
Authors have addressed reviewer comments and manuscript has significantly improved.
Scientific name in tittle is still not in italics.
i would recommend a final English editing.
Author Response
Dear reviewer,
Thank you very much for your comments and professional advice. These opinions help to improve academic rigor of our article. Based on your suggestion and request, we have made corrected modifications on the revised manuscript. First of all, we apologize for our oversight in not italicizing the scientific name of Partridge Tea, and we have corrected this error (line 2 - 3, line 26). Then, we have also made appropriate changes for the language quality of the manuscript. We hope that our work can be improved again. Finally, we hope that our revisions this time will meet the publication requirements of your journal. Thank you again for taking time out of your busy schedule to review our manuscript.
Thank you very much for your attention and time.
Your sincerely.
Xinxin Gui
Reviewer 3 Report
The improvement in the document adjusted by the authors is noticeable. Only some final recommendations.
The scientific name of the plant (in the title of the manuscript) should be in italics.
Please correct the mass range in line 193, I would think it should be 50-550 m/z and not 50-5500 m/z as reported there.
Reference 24 should be corrected:
Carazzone, C.; Rodríguez, J.P.G.; Gonzalez, M.; López, G.-D. Volatilomics of Natural Products: Whispers from Nature. In Metabolomics-Methodology and Applications in Medical Sciences and Life Sciences; IntechOpen: London, UK, 2021; Volume 11, p. 13, ISBN 0000957720.
Author Response
Dear reviewer,
Thank you very much for your comments and professional advice. These opinions help to improve academic rigor of our article. Based on your suggestion and request, we have made corrected modifications on the revised manuscript. We hope that our work can be improved again. Furthermore, we would like to show the details as follows:
Comment 1: The scientific name of the plant (in the title of the manuscript) should be in italics.
Response: Thank you for your comments and I apologize for our oversight! We have set the scientific name of Partridge Tea in italics. (line 2 - 3, line 26)
Comment 2: Please correct the mass range in line 193, I would think it should be 50-550 m/z and not 50-5500 m/z as reported there.
Response: Thank you for your comments and I apologize for our oversight! We have corrected 50-5500 m/z in the manuscript to 50-550 m/z. (line 196)
Comment 3: Reference 24 should be corrected: Carazzone, C.; Rodríguez, J.P.G.; Gonzalez, M.; López, G.-D. Volatilomics of Natural Products: Whispers from Nature. In Metabolomics-Methodology and Applications in Medical Sciences and Life Sciences; IntechOpen: London, UK, 2021; Volume 11, p. 13, ISBN 0000957720.
Response: Thank you for your comments and I apologize for our oversight! We have corrected Reference 24 in the manuscript to Carazzone, C.; Rodríguez, J.P.G.; Gonzalez, M.; López, G.-D. Volatilomics of Natural Products: Whispers from Nature. In Metabolomics-Methodology and Applications in Medical Sciences and Life Sciences; IntechOpen: London, UK, 2021; Volume 11, p. 13, ISBN 0000957720. (line 316 - 318)
Thank you very much for your attention and time.
Your sincerely.
Xinxin Gui